# Discrepancies in Electronic Medical Prescriptions Found in a Hospital Emergency Department: A Prospective Observational Study

**DOI:** 10.3390/ph17040460

**Published:** 2024-04-03

**Authors:** David García González, Paulo Teixeira-da-Silva, Juan José Salvador Sánchez, Jesús Ángel Sánchez Serrano, M. Victoria Calvo, Ana Martín-Suárez

**Affiliations:** 1Pharmaceutical Sciences Department, Universidad de Salamanca, 37007 Salamanca, Spain; toyi@usal.es (M.V.C.); amasu@usal.es (A.M.-S.); 2Pharmacy Service, León University Healthcare Complex, 24008 Leon, Spain; 3Institute of Biomedical Research of Salamanca (IBSAL), 37007 Salamanca, Spain; 4Emergency Department, Salamanca University Healthcare Complex, 37007 Salamanca, Spain; jjsalvador@saludcastillayleon.es (J.J.S.S.); jasanchezs@saludcastillayleon.es (J.Á.S.S.)

**Keywords:** shared medication record, medication reconciliation, electronic prescribing system, clinical pharmacist, emergency department

## Abstract

The medication in an electronic prescribing system (EPS) does not always match the patient’s actual medication. This prospective study analyzes the discrepancies (any inconsistency) between medication prescribed using an EPS and the medication revised by the clinical pharmacist upon admission to the observation area of the emergency department (ED). Adult patients with multimorbidity and/or polypharmacy were included. The pharmacist used multiple sources to obtain the revised medication list, including patient/carer interviews. A total of 1654 discrepancies were identified among 1131 patients. Of these patients, 64.5% had ≥1 discrepancy. The most common types of discrepancy were differences in posology (43.6%), commission (34.7%), and omission (20.9%). Analgesics (11.1%), psycholeptics (10.0%), and diuretics (8.9%) were the most affected. Furthermore, 52.5% of discrepancies affected medication that was high-alert for patients with chronic illnesses and 42.0% of medication involved withdrawal syndromes. Discrepancies increased with the number of drugs (ρ = 0.44, *p* < 0.01) and there was a difference between non-polypharmacy patients, polypharmacy ones and those with extreme polypharmacy (*p* < 0.01). Those aged over 75 years had a higher number of prescribed medications and discrepancies occurred more frequently compared with younger patients. The number of discrepancies was larger in women than in men. The EPS medication record requires verification from additional sources, including patient and/or carer interviews.

## 1. Introduction

Electronic prescribing systems (EPSs) are increasingly used worldwide. In Europe, each country has its own EPS, which consists of a central server database with end-user applications and different authentication procedures (e.g., smart card, identity card, national identity number, etc.) [1].

Spain is part of the European Union’s (EU) ePrescription and eDispensation “MyHealth@EU” project, which allows medication prescribed by a healthcare professional using an EPS to be dispensed in other EU countries [2,3]. The EU estimates that the process will be completed by the end of 2025. According to Bruthans et al., 2023, “although MyHealth@EU cross-border services have great potential to launch true patient-centered healthcare across the EU, where medical data related to travel with/for EU patients, widespread use will still take some time” [3]. The latest update on the progress of the “MyHealth@EU” project in the EU report is that it is operational in nine countries (Cyprus, the Czech Republic, Estonia, Finland, Greece, Latvia, Poland, Portugal and Spain), with the remainder in the development or testing phase [2].

The Spanish EPS is managed by the National Health System and is a digital healthcare support service that allows physicians to issue and transfer prescriptions using electronic means so that they can later be dispensed by any pharmacy upon presenting the patient’s personal healthcare card [4].

The EPS has advantages for physicians, pharmacists, patients, and the healthcare system itself [5]. It is a fast source of information on the medication that a patient has been prescribed by different healthcare professionals, which is especially useful in the Emergency Department (ED), where it is not always possible to obtain such information from the patient [6]. The main drawbacks of this tool seem to be associated with its use, due to time constraints as regards revising information and removing duplications or erroneous records [5].

A cooperative action known as the European Union Network for Patient Safety and Quality of Care (PaSQ) was launched in 2012 to promote patient safety in the European Union and facilitate the exchange of experiences between Member States and other organizations interested in aspects related to quality of care and patient safety [7]. As part of this initiative, medication reconciliation and medication review in the ED performed by a clinical pharmacist was implemented in our hospital. A recently published positioning document on the role of the clinical pharmacist in Spanish EDs establishes the basis and develops a framework of basic and advanced activities in this healthcare setting [8]. The document includes 25 activities grouped into five categories: logistics management, pharmaceutical care, risk management, training, and research. This will be gradually implemented according to the availability of resources. Among the pharmaceutical care activities, medication reconciliation and thorough medication review stand out.

According to the WHO, medication reconciliation is the formal process in which healthcare professionals partner with patients to ensure accurate and complete medication information transfer at interfaces of care [9]. This activity involves creating a patient’s Best Possible Medication History (BPMH) after a care transition and comparing that list to the prescriber’s orders. The differences between the two lists are called discrepancies and should be discussed with the prescriber to assess their justification and be corrected if required. The changes made must be documented and adequately communicated to the patient’s next healthcare manager as well as to the patient [10]. The percentage of discrepancies found in medication reconciliation and their potential impact on the patient’s health depends on numerous factors such as age, clinical condition, or the number of medications that he or she is taking. Although discrepancies are observed in the vast majority of the treatments used by polypharmacy patients, not all of them are medication errors [11,12,13]. Between 10% and 67% of medication histories have at least one error, and up to 33% of these errors have the potential to cause patient harm [14]. While there is currently no consensus regarding the definition of polypharmacy, the term most commonly refers to the use of five or more medications daily [15]. Medication errors are defined as preventable events that may cause or lead to inappropriate medication use or patient harm while the medication is in the control of the healthcare professional, patient, or consumer [16]. Such errors may cause harm, additional costs, and even death.

Medication review is a thorough and structured assessment of a patient’s medication with the purpose of identifying and addressing issues in order to improve his or her health [17]. This process usually involves tools such as criteria to identify potentially inappropriate prescriptions, especially in the case of older adults, with the Beers criteria being the most widely used in the USA and the STOPP-START in Europe [18,19]. These potentially inappropriate prescriptions are common (16.0–20.8% upon admission in patients aged ≥65 years) [20] and entail an important threat to patient safety, thus increasing healthcare costs [21]. Several studies and meta-analyses have brought to light the positive results of clinical pharmacists’ work in the management of medication reconciliation and review in EDs [22,23,24,25].

So that clinical pharmacists can perform such activities in the ED, it is necessary to know whether the EPS is a reliable source of information on a patient’s current medication list. However, as far as we know, the few studies that have analyzed the differences between what is registered on the EPS and patients’ current medication [17,26,27,28] cannot be fully extrapolated to our case because, in addition to using different EPSs, they analyzed data from a small number of patients of different ages and number of drugs used, and in a variety of hospital units. This is why we propose this work, where the term discrepancy stands for the difference between the medication recorded on the EPS and patients’ current medication, by analogy with the medication reconciliation process.

The aim of this study is to assess the correspondence of the EPS record to patients’ current treatment, identifying the therapeutic groups that are most involved in discrepancies, and analyzing certain factors associated with the appearance of discrepancies.

## 2. Results

### 2.1. Patient Characteristics

Table 1 shows certain characteristics of the 1131 patients included in the study. Most of them were older adults (65.0% of patients were over 75 years old) and had polypharmacy (75.8% of patients had five or more prescribed medications on list 2). Women’s ages were statistically higher than men’s (mean (SD) [interval]: 78.7 (12.15) [30–102] vs. 75.6 (11.7) [30–100], *p* < 0.001). There was a larger percentage of polypharmacy patients among those over 75 years old (82.4 vs. 63.4, *p* < 0.001).

### 2.2. Prescribing Discrepancies

A total of 1654 discrepancies were found. There was at least one discrepancy in 64.5% (729) of patients. The median number of discrepancies per patient was 1 (interquartile range: 0–2). Figure 1 shows the distribution of discrepancies per patient.

### 2.3. Types of Prescribing Discrepancies

Table 2 shows the frequency of occurrence of each type of discrepancy, the most commonly identified being different dosage, frequency, or route of administration. Of these types of discrepancies, 18.6% corresponded to drugs prescribed on a regular basis when their use is on demand. One out of every two discrepancies affected drugs that are considered high-alert medications for patients with chronic illnesses (HAMC) and two out of every five discrepancies involved drugs associated with withdrawal symptoms.

Table 3 shows the therapeutic groups that were the most involved in discrepancies, analgesics being the most frequent.

### 2.4. Factors Associated with Prescribing Discrepancies

As shown in Table 4, there were more patients with discrepancies in the over-75-year-old group (*p* < 0.01)—who also had a larger number of prescribed drugs (*p* < 0.01)—than in the younger group. Therefore, being over 75 years old is a risk factor for discrepancies (OR: 1.5 [1.17–1.94] *p* < 0.05). However, there was no significant difference in the number of discrepancies detected between the two age groups.

In the comparison of data between women and men, women yielded a larger number of discrepancies (*p* < 0.05), although the difference in the number of prescribed drugs was not significant and neither was the percentage of patients with discrepancies (OR: 1.21 [0.95–1.55]) (*p* > 0.05).

According to the level of polypharmacy (non-polypharmacy, polypharmacy and extreme polypharmacy), there was a significant difference between the percentage of patients with discrepancies and the number of discrepancies observed for the patients in each group (*p* < 0.01). There is a correlation between the number of drugs and the number of discrepancies (ρ = 0.44, *p* < 0.01), as well as between age and the number of discrepancies (ρ = 0.10, *p* < 0.01). The analysis of the differences between patients with different levels of polypharmacy shows that there are significant differences associated with age in the three groups (*p* < 0.01); specifically, non-polypharmacy patients are significantly younger (*p* < 0.01). There are no significant differences regarding the percentage of women among the three groups (*p* > 0.05).

## 3. Discussion

To ensure the safety of patients who attend the ED department, it is essential to have accurate and thorough information about their pharmacological therapy. Unfortunately, there is no single, corroborated source with a record of all the patient’s medication. In our country, the EPS has been implemented for the electronic transfer of prescriptions. The general practitioner in charge of monitoring the patient’s therapy must ensure that the treatments are updated, although this does not exempt the rest of specialist care physicians from updating the EPS with any changes in therapy that they may make.

According to our study, only 35.5% of patients showed consistency between the medication listed on the EPS records and the one they actually used. Not having an accurate medication history can lead to prescribing errors and mistakes in clinical decision-making, resulting in undesired effects on patients’ health [17].

### 3.1. Results in the Context of Other Studies

Studies conducted in other countries also reveal that national dispensation data repositories fail to provide an accurate picture of the medication used by patients, even if the platform can be accessed by prescribers from different healthcare levels [17,26,27,28]. There is a wealth of literature on the discrepancies detected in medication reconciliation across transitions in healthcare, but discrepancies between the EPS and patients’ current medication are scarcely addressed. Furthermore, the comparison of results is hindered by the use of different terminology for the types of discrepancies.

In Denmark, three studies that also analyze discrepancies in the EPS upon access to the ED have been conducted [26,27,28]. In our study, discrepancies were less frequent (median of one discrepancy per patient) than those reported in the mentioned studies, which was between two and three discrepancies per patient [26,27,28]. This can be explained by the different methodologies used in each study, since some of them only included patients who were using five or more prescribed drugs. The fact that in our study the patients had a smaller number of prescribed medications could be used to justify that the number of patients found in our study with at least one discrepancy (64.5%) was lower than that reported in the literature (75% [26], 78% [29], 81% [27], 88% [28] y 99% [17]). However, patients with extreme polypharmacy (group C) have a median of two discrepancies per patient, which is similar to the results of these studies [26,27,28], since the average number of medications prescribed to these patients was larger than 10. According to our study, a larger number of prescribed medications is associated with a greater risk for discrepancies, a finding that is consistent with other studies [26,27].

The therapeutic groups where more discrepancies were identified were analgesics, psycholeptics, diuretics, antidiabetics, and anti-inflammatory drugs, which concurs with the results of other studies [26,27]. However, in our case, analgesics and anti-inflammatory drugs were found to be associated with a higher number of discrepancies, probably because over-the-counter (OTC) medication indicated by the physician was also considered. It should be noted that in our country OTC medications are not financed by the public system even though they must be included in the EPS. Yet, for different reasons, the latter is not always done.

The type of discrepancy that was most frequently detected was related to dosage (43.6%) followed by commission (34.7%), as also reported in a study carried out in a similar context to ours [29]. In their study, Elliott et al. also reported these two discrepancies as the most frequent, although in the reverse order. The commission discrepancy was also the most common in other studies [26,27]. The third most common discrepancy was that of omission (20.9%), which is particularly important in medications that produce withdrawal symptoms [30]. The least common discrepancy is associated with situations where the patient mistakenly used a different drug from the one on the electronic prescription (0.7%).

The patients over 75 years old who participated in the study had a larger number of prescribed medications than the younger ones and were, therefore, more likely to have discrepancies. Although there is a correlation between age and the number of discrepancies (ρ = 0.104, *p* < 0.01), however, the total number of discrepancies per patient was not significantly higher when the participants over 75 years old were compared with the younger participants. Women yielded a larger number of discrepancies per patient than men (*p* < 0.05), although no differences could be demonstrated regarding the number of drugs or the percentage of patients with discrepancies. In their study, Bülow et al., who also analyzed possible factors related to the rate of discrepancies, found no association between the number of discrepancies per medication and sex or increasing age [26]. On the other hand, Andersen et al. reported that patients over the age of 65 had reduced rates of discrepancies per medication compared with patients aged <65 years old [28]. There are also authors who mention other factors, such as a recent revision of the EPS list or assistance in the dispensing of their medications, which could reduce the rate of discrepancies [26,28].

The differences in results in published studies with similar aims to ours may stem from the fact that they have been conducted with data collected from a single center, using small samples of patients with different characteristics, as well as from the influence of the different ways in which prescribers act. This implies the need to perform multicentric studies with a large number of patients. Among the findings, attention should be drawn to the large percentage of medications that are HAMC (52.5%) and of those associated with withdrawal symptoms (42.0%) that are involved in the discrepancies detected (Table 2). This increases the risk of adverse effects of the medication on the patient’s health [30,31].

The results of our study bring to light the safety problem that relying only on the EPS as a source of medication information poses for the patient, as corroborated by other studies [32,33,34,35]. All the healthcare experts treating a patient should update the EPS each time there is a change in therapy; however, this is not always done [28]. In practice, keeping an accurate medication record requires considerable time [22,36], as well as training [37]. To achieve a correct medication history involves consulting at least two different sources of information [38,39]. As regards this aspect, clinical pharmacists play a crucial role since they can actively discuss the use of medication with the patient/carer [40].

The “MyHealth@EU” program is expected to become gradually implemented until full operability is reached in 2025 [41]. This will be a further step in the integration of Health in the European Union (EU), allowing community pharmacists to dispense medication regardless of the country where it was prescribed.

### 3.2. Strengths and Limitations

Few studies assess the accuracy of the EPS as a source of information about a patient’s treatment, which highlights the need for studies such as this one that review the platforms used in different countries and clinical environments. The main strength of this study is that it identifies the daily clinical challenge posed in the ED by discrepancies between the EPS list and patients’ actual medication use. The number of patients included in the study is considerably larger than in other published works. Furthermore, it provides an analysis of the impact of discrepancies on medications that are classified as HAMC and on medications associated with withdrawal symptoms.

Among the limitations of the study is the fact that it was conducted in a single center and results are not necessarily generalizable to other healthcare settings. There was only one pharmacist working on the project, so patients were only assessed during his working hours, which were mornings from Monday to Friday, involving a heavy workload. The timing of the study could also be considered a limitation, as there may be variations in the frequency of discrepancies found on different days of the week. Furthermore, our results also depend on the accuracy of patients’ reported use of medication. The study was not designed to investigate the clinical significance or long-term consequences of prescribing discrepancies and, therefore, we could not evaluate the effectiveness of a pharmacist-based intervention.

## 4. Materials and Methods

### 4.1. Setting

The study was conducted in the University Healthcare Complex of Salamanca, which serves a patient population of around 320,000 inhabitants. Each year, approximately 150,000 patients resort to its ED, with a mean of 420 people a day [42,43].

### 4.2. Ethics Approval

Data collection was performed during routine patient care in the ED. The study was approved by the ethics committee of the center (protocol code CEIm: PI 2021 05 793). All data were stored anonymously.

### 4.3. Design and Patients

This prospective observational study was conducted with data from patients with multimorbidity and/or polypharmacy older than 18 years old admitted to the ED observation area of the University Hospital of Salamanca from February to June 2018.

The clinical pharmacist recruited to perform the PaSQ project’s medication reconciliation and revision functions [7] interviewed patients/carers from the ED observation department on workdays, between 08:00 and 15:00, to obtain the home medication list.

The exclusion criteria were lack of access to the patient’s EPS record and lack of information regarding pharmacological therapy on the part of the patient or carer.

Multimorbidity was considered as the presence of two or more chronic pathologies [44], polypharmacy as the use of five or more scheduled or on-demand medications [15], and the prescription of more than 10 drugs was defined as extreme polypharmacy [45]. Phytotherapy and homeopathy were not considered in the study. All the prescribed medicines were coded under level 2 of the ATC classification system.

### 4.4. Data Collection and Variables

To obtain the patient’s BPMH, the clinical pharmacist interviewed the patients/carers in the ED observation area, revising the prescriptions that were active on the EPS (list 1), primary and specialist care medical histories, or socio-medical center reports. The information gathered was used to draw up the revised list of medications (list 2) for each of the patients who met the inclusion criteria, including for each pathology the medication used, route of administration, dosage, and frequency.

Subsequently, the clinical pharmacist compared the two medication lists and the differences found were identified as discrepancies and reported to the attending physician alongside other drug-related problems detected.

For each patient, the clinical pharmacist recorded in the study’s data collection logbook the age, sex, number of prescribed drugs, number of discrepancies, types of discrepancies, and medications affected by the discrepancy.

### 4.5. Outcomes

The primary outcome was the number and type of discrepancies found, defining discrepancy as any inconsistency between the medication prescribed on the EPS (list 1) and the medication recorded by the clinical pharmacist (list 2). Discrepancies were classified as (a) commission, (b) different dosage, route, or frequency, (c) omission, or (d) wrong drug (different drug, but same therapeutic group). The patient/carer’s unilateral decision to take/deliver the medication in a different way from that prescribed by the physician was not considered a discrepancy. OTC medications not subjected to medical opinion were not considered, because the study is aimed at measuring consistency between patients’ prescriptions and what is registered in the EPS.

Secondary outcomes were the analysis of the therapeutic groups (using the ATC classification code) and factors associated with the discrepancies.

To assess the potential relevance of the discrepancies, the presence of prescribed drugs that were regarded as HAMC was considered (Table 5), which are medications that are highly likely to cause severe harm or even death if an error occurs in the course of their use [46], as well as drugs that involve withdrawal symptoms (Table 6), which are those whose sudden interruption can cause undesirable or rebound effects [47].

### 4.6. Statistics

The statistical analysis was performed using IBM SPSS Statistics^®^ version 25.0 software. For all the tests, the statistical significance threshold was set at *p* < 0.05. Qualitative variables were expressed as absolute frequencies and percentages, quantitative ones as mean and standard deviation if the variable followed a normal distribution, or median and interquartile range if otherwise. The comparison between quantitative variables was performed using Student’s *t*-test, ANOVA and Kruskal–Wallis tests; the post-hoc comparison was performed using the Tukey method.

The association between age and sex with the presence or absence of discrepancies was expressed using the odds ratio (OR) and confidence intervals (CI 95%). The correlation between the number of drugs and age vs. the number of discrepancies was assessed using Spearman’s rank-order correlation.

## 5. Conclusions

Even though obtaining accurate patient medication information upon admission to the ED is challenging, it is essential for patient safety. Out of 1131 patients admitted to the ED, 64.5% did not have their prescribed medication correctly updated on the EPS, with a median of one discrepancy between the EPS and patients’ actual use of medication. The most common discrepancy was associated with dosage; the most involved medications were analgesics, neuroleptics, and diuretics, most of which are classified as HAMC. The increase in the number of prescribed drugs was associated with a higher frequency of discrepancies. These findings show that this source of information on patients’ prescribed medication requires verification using additional sources, including interviews with patients and/or their careers.

## Figures and Tables

**Figure 1 pharmaceuticals-17-00460-f001:**
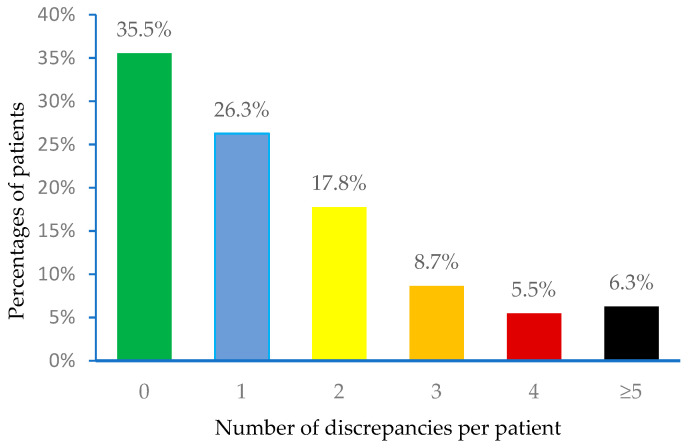
The percentage of patients with discrepancies found between the medication prescribed on the electronic prescribing system (list 1) and the medication revised by the clinical pharmacist (list 2).

**Table 1 pharmaceuticals-17-00460-t001:** Characteristics of the patients included in the study and prescribed drugs.

Patients, *n*	1131
Sex (Female/Male), *n* (%)	591 (52.3)/540 (47.7)
Age, years (mean (SD) (interval))	78.0 (11.8) (30–102)
Prescribed drugs, *n*	9238
Prescribed drugs/patient, *n* (mean (SD) (interval))	8.2 (3.9) (2–22)

*n*, number; SD, standard deviation.

**Table 2 pharmaceuticals-17-00460-t002:** Types of discrepancies found.

Type of Discrepancy	Discrepancies, *n* (%)	HAMC, *n* (%)	DWS, *n* (%)
Different DFR	721 (43.6)	446 (61.9)	367 (37.0)
Commission	575 (34.7)	265 (46.1)	209 (36.3)
Omission	346 (20.9)	151 (43.6)	111 (32.0)
Wrong drug	12 (0.7)	6 (50.0)	7 (58.3)
Total	1654	868 (52.5)	694 (42.0)

*n*, number; HAMC, high-alert medications for patients with chronic illnesses; DWS, drug associated with withdrawal symptoms; DFR, dosage, frequency, or route.

**Table 3 pharmaceuticals-17-00460-t003:** Frequency of discrepancies found according to therapeutic group.

ATC-Drug Group (Level 2)	Description	Number of Discrepancies, *n* (%)
N02	Analgesics	183 (11.1)
N05	Psycholeptics	165 (10.0)
C03	Diuretics	147 (8.9)
A10	Drugs used in diabetes	110 (6.7)
M01	Anti-inflammatory and antirheumatic products	79 (4.8)
A02	Gastric mucosa protective agents	74 (4.5)
B01	Antithrombotic agents	71 (4.3)
C07	Beta-blockers	71 (4.3)
C09	Agents acting on the renin–angiotensin system	66 (4.0)
R03	Drugs for obstructive airway disease	60 (3.6)
J01	Antibacterials for systemic use	53 (3.2)
A12	Trace elements	47 (2.8)
H02	Corticosteroids for systemic use	45 (2.7)
A06	Laxatives	43 (2.6)
C01	Antiarrhythmics	41 (2.5)
N06	Phsychoanaleptics	39 (2.4)
C10	Hypolipidemic agents	39 (2.4)
	Others Group A	72 (4.4)
	Others Group N	68 (4.1)
	Others Group C	63 (3.8)
	Others Group M Rest of groups	25 (1.5) 97 (5.9)

*n*, number of discrepancies; Group A, digestive system drugs Group N, central nervous system drugs; Group C, cardiovascular drugs; Group M, musculoskeletal system drugs.

**Table 4 pharmaceuticals-17-00460-t004:** Factors that could influence the appearance of discrepancies between the medication prescribed on the electronic prescribing system (list 1) and the medication revised by the clinical pharmacist (list 2).

	Patients, *n* (%)	Patients with Discrepancies, *n* (%)	Drugs per Patient, Median (Interquartile Range)	Discrepancies per Patient, Median (Interquartile Range)
Polypharmacy				
Group A (≤4)	274 (24.2)	* 128 (46.7)	* 4 (3–4)	* 0 (0–2)
Group B (5–9)	534 (47.2)	* 350 (65.5)	* 7 (6–8)	* 1 (0–2)
Group C (≥10)	323 (28.6)	* 259 (80.2)	* 12 (11–15)	* 2 (1–4)
Age				
<75 years old	396 (35.0)	* 232 (58.6)	* 4 (5–9)	1 (0–2)
>75 years old	735 (65.0)	* 500 (68.0)	* 8 (5–11)	1 (0–2)
Sex				
Male	540 (47.7)	337 (62.4)	8 (5–10)	** 1 (0–2)
Female	591 (52.3)	395 (66.8)	8 (5–10)	** 1 (0–2)

Group A: non-polypharmacy, Group B: polypharmacy, Group C: extreme polypharmacy. * *p* < 0.01; ** *p* < 0.05.

**Table 5 pharmaceuticals-17-00460-t005:** Drugs regarded as high-alert medication for patients with chronic illnesses [46].

High-Alert Therapeutic Groups	
**Antiplatelets**	**Beta-Adrenergic Blockers**
Oral anticoagulants	Oral cytostatics
Narrow therapeutic index antiepileptic drugs	Immunosuppressors
NSAIDs	Loop diuretics
Antipsychotics	Oral hypoglycemic agents
Benzodiazepines and similar Insulins	Corticosteroids used long-term (more than 3 months)
Other Specific High-Alert Drugs	
Amiodarone/Dronedarone	Spironolactone/Eplerenone
Digoxin	Oral methotrexate (non-cancer use)

NSAID, Nonsteroidal anti-inflammatory drug.

**Table 6 pharmaceuticals-17-00460-t006:** Drugs involving withdrawal symptoms [47].

Antihypertensives *	Beta-Blockers
Antidepressants	Opioids
Antipsychotics	Corticoids
Antiparkinsonians	Inhalers
Antiepileptic drugs	Proton pump inhibitors
Anti-Alzheimer drugs	Nitrates
	Methylphenidate/atomoxetine

* Diuretics used as antihypertensives are excluded.

## Data Availability

Data are contained within the article.

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
