# Peer review of "Discrepancies in Electronic Medical Prescriptions Found in a Hospital Emergency Department: A Prospective Observational Study"

_pharmaceuticals, 2024, doi:10.3390/ph17040460_

Round 1
Reviewer 1 Report
Comments and Suggestions for Authors
Page 2, Line 55 - The use of "gathers" is not appropriate in this setting.
Kindly replace with "Table 1 shows"
Page 2,Table 1 - The right nomenclature for "Man/Woman" is "Male/Female". Kindly revise.
Page 2, Line 64 - Mean 1.5 (SD 1.6). Having the SD>mean confirms that the data is not normally distributed. in this case the use of median as a measure of central tendency is standard. Kindly revise.
Page 3, Line 87 - "table 3". Please use "Table" throughout the manuscript for consistency
Comments on the Quality of English Language
The standard of the language is good apart from the highlighted comments
Author Response
Response to general comments. We would like to thank the reviewer for the positive overall feedback. We consider the comments and suggestions valuable and relevant to help improve the manuscript. For better readability, we have repeated each of the reviewer’s comments before providing our answer. As far as possible, we have paid attention to all the grammatical, orthographic and statistical errors detected. The line numbers in our answers refer to those in the revised manuscript. Text that has been added to or removed from the manuscript can be seen using “track changes”. We hope that the new version of the manuscript now qualifies for publication in the journal Pharmaceuticals.
Point 1. Page 2, Line 55 - The use of "gathers" is not appropriate in this setting. Kindly replace with "Table 1 shows"
Response 1. We appreciate the reviewer's suggestion and have made the proposed change.
Point 2. Page 2, Table 1 - The right nomenclature for "Man/Woman" is "Male/Female". Kindly revise.
Response 2. We appreciate the reviewer's suggestion and have made the proposed change.
Point 3. Page 2, Line 64 - Mean 1.5 (SD 1.6). Having the SD>mean confirms that the data is not normally distributed. in this case the use of median as a measure of central tendency is standard. Kindly revise.
Response 3. We agree with the reviewer and apologize for this mistake. We have replaced it with median and interquartile range. Because of this, in the statistics section, we have replaced Pearson's correlation with Spearman's correlation coefficient (line 348). This test proved a correlation between the number of medications and age and a greater number of discrepancies. Furthermore, we have modified lines 22-25, 159-160, 187-188, 219-220, and 351 to adapt them to the referee’s suggestions.
In Table 4, we have replaced mean, standard deviation, and interval with median and interquartile range, in keeping with the reviewer's suggestions, because there is a non-normal distribution. In lines 120-121, we have replaced mean and standard deviation with median and interquartile range.
Point 4. Page 3, Line 87 - "table 3". Please use "Table" throughout the manuscript for consistency
Response 4. We appreciate the reviewer's suggestion. As the reviewer correctly pointed out, we have checked and corrected these possible orthographic errors (lines 146,239, 326, 329).

Reviewer 2 Report
Comments and Suggestions for Authors
This manuscript describes a prospective observational study to assess medication discrepancies between an electronic prescribing system and how patients were actually taking medications. The topic is of importance in pharmacy practice (although the connection to pharmaceuticals and pharmaceutical science is less clear). However, there is an overall lack of clarity in many aspects of the manuscript and a larger problem related to the contextualization of the study within the broader literature base. Specific comments are noted below.
Introduction
1) This section is much too brief and needs to address the following: a) what is currently known about medication discrepancies prior to the introduction of the EPS?; b) what other studies have been conducted on the EPS related to medication discrepancies or related areas? (this should be both within Spain and from other applicable countries); c) what is the literature and/or evidence gap being addressed? (some of this is in the Discussion but should be moved here instead); d) what is the role of the pharmacist related to both medication reconciliation to address discrepancies and with the EPS specifically?; and e) besides education transfer, does the EPS populate the receiving electronic system with the medication information in the patient's profile (e.g. similar to electronic prescribing and electronic health records in the United States)?
Discussion
2) Lines 127-129: So how does the current study's sub-analysis of just groups B and C compare with the existing literature?
3) Line 138: Inclusion of OTC medications is not explicitly mentioned in the Materials and Methods and should be, along with a justification for inclusion. As the focus of the current study is on the EPS, this means that prescribed medications should be the sole type of medication examined (unless medical practice in Spain is such that OTC medications get prescribed through the EPS - but this is also something that is not described in the Introduction).
4) Lines 148-160: What do the authors believe the implications are for the varied findings across studies? Don't just note that there are differences, provide potential reasons or note if future studies are needed.
5) Lines 176-178: Is there no current estimate of EPS adoption? If yes, that might be important contextual information to be included in the Introduction.
Materials and Methods
6) Line 211: Revise medication for what purpose? Be specific.
7) Lines 215-216: Awkward wording - is it 5 or more medications that are any combination of "established regimen" (which I assume is the same as a scheduled medication), on-demand, or rescue (which should be the same as on-demand as it should only be used in emergent situations).
8) Lines 222-223: Consulting them for what purpose? Again, be specific.
9) Lines 236-238: Why were these discrepancy classifications chosen?
10) Lines 257-258: If a variable followed a non-normal distribution, what statistical test was used?
Comments on the Quality of English LanguageSome minor edits needed, but English is generally fine.
Author Response
General comments. This manuscript describes a prospective observational study to assess medication discrepancies between an electronic prescribing system and how patients were actually taking medications. The topic is of importance in pharmacy practice (although the connection to pharmaceuticals and pharmaceutical science is less clear). However, there is an overall lack of clarity in many aspects of the manuscript and a larger problem related to the contextualization of the study within the broader literature base. Specific comments are noted below.
Response to general comments.
We would like to thank the reviewer for the overall positive feedback. We consider the comments and suggestions valuable and relevant to help improve the manuscript. We have tried to clarify and contextualize the study according to your suggestions. As the reviewer says, this topic is of importance in pharmacy practice, which is why the journal Pharmaceuticals publishes a special issue related to this topic. For better readability, we have repeated each of the reviewer's comments before providing our answer. Text that has been added to or removed from the manuscript can be seen using “track changes”. As far as possible, we have paid attention to all the grammatical errors detected. The final version of the manuscript has been carefully revised by a professional native English editor with experience in the area. We hope that the new version of the manuscript is now to your liking.
Introduction
Point 1. This section is much too brief and needs to address the following:
- a) what is currently known about medication discrepancies prior to the introduction of the EPS?; b) what other studies have been conducted on the EPS related to medication discrepancies or related areas? (this should be both within Spain and from other applicable countries); c) what is the literature and/or evidence gap being addressed? (some of this is in the Discussion but should be moved here instead); d) what is the role of the pharmacist related to both medication reconciliation to address discrepancies and with the EPS specifically?; and e) besides education transfer, does the EPS populate the receiving electronic system with the medication information in the patient's profile (e.g. similar to electronic prescribing and electronic health records in the United States)?
Response 1. We agree with the reviewer that the introduction was too brief. We have added 4 paragraphs (lines 58-104) to clarify the different aspects mentioned by the reviewer. Data on what discrepancies and medication errors entail have been included. We have attempted to clarify what is considered a discrepancy in medication reconciliation and the EPS discrepancies that are the subject of our study. Other studies conducted on the EPS related to medication discrepancies have been referenced. Likewise, an attempt has been made to explain the role of the pharmacist in the identification of discrepancies and the need to know the reliability of the EPS as a source of information on the patient's medication.
Discussion
Point 2. Lines 127-129: So how does the current study's sub-analysis of just groups B and C compare with the existing literature?
Response 2.
Thank you very much for your suggestion. We fully agree with it and have added, in lines 191-194, a sub-analysis of our patients with extreme polypharmacy (group C) and the patients from these studies since they had a similar average number of medications prescribed.
Point 3. Line 138: Inclusion of OTC medications is not explicitly mentioned in the Materials and Methods and should be, along with a justification for inclusion. As the focus of the current study is on the EPS, this means that prescribed medications should be the sole type of medication examined (unless medical practice in Spain is such that OTC medications get prescribed through the EPS - but this is also something that is not described in the Introduction).
Response 3. We agree with the reviewer. Indeed the use of OTC medications was not explicitly mentioned in the study. For this reason, we have added lines 204 to 207 in the Discussion section, and lines 320-322 in the Materials and Methods section.
In our country, OTC medications are not financed by the public system even though they should be included in the EPS. Therefore, in our study, we have only considered OTC medications indicated by the physician. We have not taken into account those taken by the patient’s own choice because the objective of this study is to measure consistency between the medication that is recorded in the EPS and that prescribed by the physician.
Point 4. Lines 148-160: What do the authors believe the implications are for the varied findings across studies? Don't just note that there are differences, provide potential reasons or note if future studies are needed.
Response 4. We appreciate the reviewer's suggestion, since it was barely explained in the paper. We have added a more extensive explanation in lines 232-236.
Point 5. Lines 176-178: Is there no current estimate of EPS adoption? If yes, that might be important contextual information to be included in the Introduction.
Response 5. Thank you very much for your suggestion, we agree with the reviewer. We have added lines 39-46 providing more information on the EU´s EPS in the Introduction section.
Materials and Methods
Point 6. Line 211: Revise medication for what purpose? Be specific.
Response 6. To clarify this point, the medication reconciliation and review activities that the pharmacist carried out in the ED have been explained in the new paragraphs of the introduction (lines 58-104). The patient's medication information obtained in these clinical activities has been used in the study to evaluate the reliability of the information recorded in the EPS. However, these activities are not the subject of our study and, therefore, to avoid confusion, we have removed "and subsequently review the medication" in lines 289-291.
Point 7. Lines 215-216: Awkward wording - is it 5 or more medications that are any combination of "established regimen" (which I assume is the same as a scheduled medication), on-demand, or rescue (which should be the same as on-demand as it should only be used in emergent situations).
Response 7. Thank you very much for your suggestion, we agree with the reviewer. The terms “regular”, “daily” and “rescue medication” in lines 295-296 have been eliminated. Now, in addition to being more accurate, it is more reader-friendly.
Point 8. Lines 222-223: Consulting them for what purpose? Again, be specific.
Response 8. We agree with the reviewer that it could lead to confusion and have changed line 303. The clinical pharmacist did not directly ask the specialist or primary care physician but instead accessed the primary care and specialized care medical records.
Point 9. Lines 236-238: Why were these discrepancy classifications chosen?
Response 9. This classification was used in our work centre, which is the one used in the PaSQ Project and where the data were collected. This classification is similar to other subsequent definitions (references 25-28). However, our definition is further simplified by unifying “Incorrect dose frequency” and “Incorrect dosage” because both refer to the patient's total daily dose and allow all discrepancies to be classified. A paragraph has also been added in the introduction (lines 58-63) commenting that the study has been carried out in the context of the PaSQ Project.
Point 10. Lines 257-258: If a variable followed a non-normal distribution, what statistical test was used?
Response 10. We appreciate this suggestion, which represents an important improvement for the study. The statistical test has been changed and Spearman's correlation coefficient has been used (line 348). This test proved a correlation between the number of medications and age and a greater number of discrepancies. Furthermore, we have modified lines 22-25, 159-160,187-188, 219-220 and 351 to adapt them to the referee's suggestions.
In Table 4, we have replaced mean, standard deviation, and interval with median and interquartile range, in keeping with the reviewer's suggestions, because there is a non-normal distribution. In lines 120-121, we have replaced mean and standard deviation with median and interquartile range.

Reviewer 3 Report
Comments and Suggestions for Authors
Review
Discrepancies in electronic medical prescriptions found in a hospital emergency department: Prospective observational study.
General:
The paper describes a significant and clinically relevant topic but needs a major revision. Many important definitions are not being explained or even mixed.
Specific comments:
Abstract
Medication reconciliation should be included and defined. Did the authors do that or check the data after the treatment?
Medication discrepancies should be defined.
INTRODUCTION
General – should be expanded significantly – no data on medication errors (main thing why we have these systems – please focus on patients).
The introduction should include “real” studies focused on the consequences of medication errors and ADRs. Fu, AZ.; Jiang, JZ.; Reeves, JH.; Fincham, JE.; Liu, GG.; Perri M 3rd. Potentially inappropriate medication use and healthcare expenditures in the US community-dwelling elderly. Med Care 2007, 45, 472-476.
Massachusetts Technology Collaborative (MTC) and NEHI, 2008. Saving Lives, Saving Money: The Imperative for CPOE in Massachusetts. Updated to 2008 figures. Cambridge, MA: NEHI, 2008. http://www.nehi.net
Makary MA, Daniel M. Medical error-the third leading cause of death in the US. BMJ. 2016 May 3;353:i2139. Doi: 10.1136/bmj.i2139. PMID: 27143499.
The introduction should also focus on inappropriate prescribing in the context of medication errors and data available on this topic:
Ay P, Akici A, Harmanc H. Drug utilization and potentially inappropriate drug use in elderly residents of a community in Istanbul, Turkey. Int J Clin Pharmacol Ther. 2005;43(4):195–202.
Egger SS, Bachmann A, Hubmann N, Schlienger RG, Krähenbühl S. Prevalence of potentially inappropriate medication use in elderly patients: comparison between general medical and geriatric wards. Drugs Aging. 2006;23(10):823–837.
Davies, EC.; Green, CF.; Mottram, DR.; Pirmohamed, M. Adverse drug reactions in hospitals: a narrative review. Curr Drug Saf 2007, 2, 79-87. Lau, DT.; Kasper, JD.; Potter, DE.; Lyles, A.; Bennett RG. Hospitalization and death associated with potentially inappropriate medication prescriptions among elderly nursing home residents. Arch Intern Med 2005, 165, 68-74.
METHODS
Methods should be given in a separate paragraph before the results.
Many mistakes are included within the methods, and few data are given (no clear how pharmacists provided recommendations, role of clinical pharmacist in a SPA system; collaboration on the ward on daily basis or not…
Medication reconciliation should be defined … not clear in this paper.
Roles and responsibilities differ among different EU countries (e.g., in the UK, clinical pharmacists can prescribe medications, same as nurses in some countries). The authors should focus on that in the bullet points given.
https://www.pharmacyregulation.org/education/pharmacist-independent-prescriber
https://human-resources-health.biomedcentral.com/articles/10.1186/s12960-019-0429-6
Statistics
Pearson’s? Authors could not know that there would be data normality …
RESULTS
Table 2 is not clear – no ORs and other variables included in the methods.
DISCUSSION
1) The authors did not mention the medication errors and events during the transition of care process (e.g., from primary care to hospital and vice versa), which should be described and mentioned. WHO has established the entire strategy for this process – medication errors minimising in context of the transition of care, which the authors did not mention: Medication Safety in Transitions of Care. Geneva: World Health Organization; 2019 (WHO/UHC/SDS/2019.9). Licence: CC BY-NC-SA 3.0 IGO.
2) In this context, they have to describe medication reconciliation at the hospital admission and discharge and seamless care process:
3) Pippins JR, Gandhi TK, Hamann C, Ndumele CD, Labonville SA, Diedrichsen EK, Carty MG, Karson AS, Bhan I, Coley CM, Liang CL, Turchin A, McCarthy PC, Schnipper JL. Classifying and predicting errors of inpatient medication reconciliation. J Gen Intern Med. 2008 Sep;23(9):1414-22. doi: 10.1007/s11606-008-0687-9. Epub 2008 Jun 19. PMID: 18563493; PMCID: PMC2518028.
4) Medication Safety in Transitions of Care. Geneva: World Health Organization; 2019 (WHO/UHC/SDS/2019.9). Licence: CC BY-NC-SA 3.0 IGO.
Please also let us know if this service is paid for in Spain. In Slovenia, clinical pharmacists provide medication reconciliation in all hospitals, and this is paid extra:
Please check the following paper:
Stuhec M, Batinic B. Clinical pharmacist interventions in the transition of care in a mental health hospital: case reports focused on the medication reconciliation process. Front Psychiatry. 2023;14:1263464.
What is about the discharge (any interventions by clinical pharmacists?)? Please check the paper above.
The authors should also describe a broader context of clinical pharmacy services in Spain (for international readers):
Pharmacists can have a significant impact on the PIMs in elderly patients. Did the authors check this issue?
Stuhec M, Zorjan K. Clinical pharmacist interventions in ambulatory psychogeriatric patients with excessive polypharmacy. Sci Rep. 2022;12(1):11387.
Limitations and strengths should be modified (e.g., STROBE, methods, no-RCT).
Author Response
General comments: The paper describes a significant and clinically relevant topic but needs a major revision. Many important definitions are not being explained or even mixed.
Response to general comments.
We would like to thank the reviewer for the overall positive feedback and interest in this research. In accordance with your comment, we have added new text in the introduction, defining and explaining the concepts considered of interest by the reviewer.
For better readability, we have repeated each of the reviewer's comments before providing our answer. The final version of the manuscript has been carefully revised by a professional native English editor with experience in the area. Text that has been added to or removed from the manuscript can be seen using “track changes”. We hope that the new version of the manuscript is now to your liking.
Abstract
Point 1. Medication reconciliation should be included and defined. Did the authors do that or check the data after the treatment?
Response 1. According to the reviewer's suggestions, we have included the definition of medication reconciliation in the introduction (lines 71-78). It has not been possible to include it in the abstract because of the limit to the number of words.
To clarify the objective of our study, we have added several paragraphs in the introduction (lines 58-63 where PaSQ is discussed). The study was carried out in the context of the PaSQ project, with the clinical pharmacist performing medication reconciliation and review. However, these activities are not the object of this investigation. The study is aimed at evaluating the reliability of the EPS as a source of information on the patient's medication. Therefore, we present data on discrepancies between the medication recorded in the EPS and the patient's current medication.
Point 2. Medication discrepancies should be defined.
Response 2. In the abstract, due to the limit to the number of words, we have only added one clarification of the term (line 12). But, according to the reviewer's suggestion, we have added a complete explanation of the concept of discrepancy in the introduction (lines 75-77).
Introduction
General – should be expanded significantly – no data on medication errors (main thing why we have these systems – please focus on patients).
General response. According to the suggestions, the introduction has been expanded to include data on medication errors and discrepancies, as well as new supporting references as proposed by the reviewer (lines71-97)
Point 3. The introduction should include “real” studies focused on the consequences of medication errors and ADRs.
- Fu, AZ.; Jiang, JZ.; Reeves, JH.; Fincham, JE.; Liu, GG.; Perri M 3rd. Potentially inappropriate medication use and healthcare expenditures in the US community-dwelling elderly. Med Care 2007, 45, 472-476.
- Massachusetts Technology Collaborative (MTC) and NEHI, 2008. Saving Lives, Saving Money: The Imperative for CPOE in Massachusetts. Updated to 2008 figures. Cambridge, MA: NEHI, 2008. http://www.nehi.net
- Makary MA, Daniel M. Medical error-the third leading cause of death in the US. BMJ. 2016 May 3;353:i2139. Doi: 10.1136/bmj.i2139. PMID: 27143499.
Point 4. The introduction should also focus on inappropriate prescribing in the context of medication errors and data available on this topic:
- Ay P, Akici A, Harmanc H. Drug utilization and potentially inappropriate drug use in elderly residents of a community in Istanbul, Turkey. Int J Clin Pharmacol Ther. 2005;43(4):195–202.
- Egger SS, Bachmann A, Hubmann N, Schlienger RG, Krähenbühl S. Prevalence of potentially inappropriate medication use in elderly patients: comparison between general medical and geriatric wards. Drugs Aging. 2006;23(10):823–837.
- Davies, EC.; Green, CF.; Mottram, DR.; Pirmohamed, M. Adverse drug reactions in hospitals: a narrative review. Curr Drug Saf 2007, 2, 79-87.
- Lau, DT.; Kasper, JD.; Potter, DE.; Lyles, A.; Bennett RG. Hospitalization and death associated with potentially inappropriate medication prescriptions among elderly nursing home residents. Arch Intern Med 2005, 165, 68-74.
Response to points 3 and 4. In accordance with these comments, several sentences have been introduced in the introduction about medication errors and potentially inappropriate prescriptions, as well as their consequences, including some of the references suggested by the reviewer (lines 78-97).
We would like to clarify that, as indicated by reviewer nº2, our manuscript describes a prospective observational study to assess medication discrepancies between an electronic prescribing system and how patients were actually taking medications. This is a subject of importance in pharmacy practice, especially in emergency services where more information about the patient is frequently not available and prompt attention is required. The objective of the study was not to analyse data on potentially inappropriate prescriptions, medication errors or adverse drug reactions.
In our study, the clinical pharmacist carefully reviewed patients' medication using the STOPP/START criteria, drug technical sheets and drug safety alerts issued by the Spanish Agency for Medicines and Health Products. Besides, all identified discrepancies and other drug-related problems were addressed together with the physician responsible for the patient. However, these data have not been included in the study because they are outside the scope of our study.
Methods
Point 5. Methods should be given in a separate paragraph before the results.
Response 5. We agree with the reviewer's proposed organization; however, the format has not been modified because it must comply with the instructions for journal authors and the template to fill out.
Point 6. Many mistakes are included within the methods, and few data are given (no clear how pharmacists provided recommendations, role of clinical pharmacist in a SPA system; collaboration on the ward on daily basis or not…
Response 6. According to the reviewer's suggestions, we have added information about the role of the clinical pharmacist in ED in the introduction (lines 64-70). The scope of activities carried out by the pharmacist is very broad, but the study’s objective is to know the degree of consistency between the EPS and the patient's actual medication in our environment, which is why we have not included data on the collaboration of the pharmacist in the adequacy of pharmacological treatments.
Point 7. Medication reconciliation should be defined … not clear in this paper.
Response 7. According to the reviewer's suggestion to clarify this point, we have included the definition of medication reconciliation in line 71 of the introduction.
Point 8. Roles and responsibilities differ among different EU countries (e.g., in the UK, clinical pharmacists can prescribe medications, same as nurses in some countries). The authors should focus on that in the bullet points given.
https://www.pharmacyregulation.org/education/pharmacist-independent-prescriber
https://humanresourceshealth.biomedcentral.com/articles/10.1186/s12960-019-0429-6
https://humanresourceshealth.biomedcentral.com/articles/10.1186/s12960-019-0429-6
Response 8. According to the reviewer's suggestions, and to facilitate understanding, we have added information on the role and responsibilities of the pharmacist (line 64). In our case, the EPS prescription was carried out only by the physician. Although the dispensing responsibilities of nurses and pharmacists are expected to increase in the future.
Point 9. Pearson’s? Authors could not know that there would be data normality …
Response 9. We agree with the reviewer. The statistics have been modified and Spearman's correlation coefficient has been used (line 348). This test proved a correlation between the number of medications and age and a greater number of discrepancies. Furthermore, we have modified lines modified 22-25, 159-160, 187-188, 219-220, and 351to adapt them to the referee's suggestions.
RESULTS
Point 10. Table 2 is not clear – no ORs and other variables included in the methods.
Response 10. Unfortunately, we do not fully understand the reviewer's suggestion regarding the information collected in the methods section. The use OR is explained in line 346, and the variables that are studied are included in lines 310-312.
Table 2 is only a descriptive table of the type and number of discrepancies. In lines 149, 153 and 154 we have completed the OR analysis according to the reviewer's suggestion.
Discussion
Point 11. The authors did not mention the medication errors and events during the transition of care process (e.g., from primary care to hospital and vice versa), which should be described and mentioned. WHO has established the entire strategy for this process – medication errors minimising in context of the transition of care, which the authors did not mention: Medication Safety in Transitions of Care. Geneva: World Health Organization; 2019 (WHO/UHC/SDS/2019.9). Licence: CC BY-NC-SA 3.0 IGO.
Response 11. We agree with the reviewer that medication reconciliation between levels of care is a highly valuable service for patient safety since care transition is where a significant number of medication errors occur. Hence, it is one of the three areas of patient safety work, as stated in the WHO technical report mentioned by the reviewer. Currently, medication reconciliation is being studied both in the hospital setting (admission, ward transition, and discharge) and community pharmacy, as reflected in many scientific works that are listed in the bibliography. However, as previously indicated, our focus is discrepancies between the medication that the patient must take and the medication collected in the EPS system, which is much less studied but equally important in the patient safety area, as it can lead to errors that have an impact on the patient's health. In line 71 of the introduction, we have added the concept of medication reconciliation and in line 100 the importance of a correctly updated EPS.
Point 12. In this context, they have to describe medication reconciliation at the hospital admission and discharge and seamless care process:
- Pippins JR, Gandhi TK, Hamann C, Ndumele CD, Labonville SA, Diedrichsen EK, Carty MG, Karson AS, Bhan I, Coley CM, Liang CL, Turchin A, McCarthy PC, Schnipper JL. Classifying and predicting errors of inpatient medication reconciliation. J Gen Intern Med. 2008 Sep;23(9):1414-22. doi: 10.1007/s11606-008-0687-9. Epub 2008 Jun 19. PMID: 18563493; PMCID: PMC2518028.
- Medication Safety in Transitions of Care. Geneva: World Health Organization; 2019 (WHO/UHC/SDS/2019.9). Licence: CC BY-NC-SA 3.0 IGO.
Response 12. We thank the reviewer for the recommendation of bibliographical references and have accordingly added the definition of medication reconciliation in line 71 of the introduction section, as well as two of the suggested references.
Point 13. Please also let us know if this service is paid for in Spain. In Slovenia, clinical pharmacists provide medication reconciliation in all hospitals, and this is paid extra: Please check the following paper:
- Stuhec M, Batinic B. Clinical pharmacist interventions in the transition of care in a mental health hospital: case reports focused on the medication reconciliation process. Front Psychiatry. 2023;14:1263464.
Response 13. In Spain, hospital care is mostly delivered in public hospitals. The activities of the clinical pharmacist depend on the person responsible for the service, hospital medical directors, hospital managers and public administration according to healthcare needs, resources, and objectives. In any case, they are free for the patient and are financed by the administration. Unfortunately, due to staff shortages and a lack of resources, there are many patients whose medication is not properly reconciled upon admission and discharge. The healthcare professionals who carry out medication reconciliation tasks receive no additional payment. Medication reconciliation performed in community pharmacies is a different matter. Some free projects have been carried out, but it is claimed that they should be financially rewarded to be viable from the community pharmacy. In lines 64-70 of the introduction section, we have added the activities that the clinical pharmacist can carry out in the emergency department.
Point 14. What is about the discharge (any interventions by clinical pharmacists?)? Please check the paper above.
Response 14. Thank you for sharing the bibliographic reference, we found it very helpful and have included it as a reference. However, as we have previously mentioned, the clinical interventions performed by the pharmacist at discharge are beyond the scope of the study. This research aims to answer a very little studied question, which is the value of EPS systems and their degree of consistency with current medication.
Point 15. The authors should also describe a broader context of clinical pharmacy services in Spain (for international readers):
Response 15. We appreciate the reviewer's suggestion and have included the main functions of the clinical pharmacist in Spain in line 64 of the introduction.
Point 16. Pharmacists can have a significant impact on the PIMs in elderly patients. Did the authors check this issue? Stuhec M, Zorjan K. Clinical pharmacist interventions in ambulatory psychogeriatric patients with excessive polypharmacy. Sci Rep. 2022;12(1):11387.
Response 16. According to the literature, potentially inappropriate prescriptions (PIPs) are associated with an increased risk of hospitalization, worsening physical function, and death in vulnerable populations, such as the elderly or patients with cognitive impairment. However, we have not included these aspects because this is not the objective of our study. Nevertheless, following this suggestion, we have added some sentences about PIPs and their clinical implications (lines 88-97), as well as the indicated reference, in the introduction.
Point 17. Limitations and strengths should be modified (e.g., STROBE, methods, no-RCT).
Response 17. We apologize to the reviewer, but we do not understand this suggestion. We have attempted, as STROBE indicates, to highlight the limitations of the study that may be sources of potential bias or imprecision. To try to clarify this section we have included lines 267-270.

Round 2
Reviewer 2 Report
Comments and Suggestions for Authors
This is a significantly improved revision, thanks to the authors for making thoughtful and responsive edits. There are just a few more minor edits to address.
Abstract
1) Should include a little bit of background about the research problem being addressed.
Introduction
2) Line 82: Polypharmacy is not previously defined – would be good to have a formal definition as it can differ.
3) Lines 100-102: Need to be more specific about why existing studies cannot be extrapolated to the current situation.
Discussion
4) Line 193: Don’t need “which report medians of 2 to 3 discrepancies per patient” as it was just noted two sentences prior and the subject did not change.
5) Line 196: Specify what “The same reason” is.
Materials and Methods
6) Line 327: Change “damage” to “harm” to be more consistent with common patient safety language.
Comments on the Quality of English LanguageEnglish language is generally fine.
Author Response
General comments. This is a significantly improved revision, thanks to the authors for making thoughtful and responsive edits. There are just a few more minor edits to address.
Response to general comments.
We would like to thank the reviewer for the overall positive feedback. We appreciate that the changes previously made were to the reviewer's liking. We consider the new comments and suggestions valuable and relevant to help improve the manuscript. We have tried to clarify and contextualize the study according to your suggestions. For better readability, we have repeated each of the reviewer's comments before providing our answer. Text that has been added to or removed from the manuscript can be seen using “track changes”. We hope that the new version of the manuscript is now to your liking.
Abstract
Point 1. Should include a little bit of background about the research problem being addressed.
Response 1. We appreciate the reviewer's suggestion and we have included some background information in line 12 of the abstract. However, this has made minor modifications to the abstract necessary due to the word limit (200). Likewise, a new sentence has been included in the conclusions section to improve it in this regard (line 349).
Introduction
Point 2. Line 82: Polypharmacy is not previously defined – would be good to have a formal definition as it can differ.
Response 2. We appreciate the reviewer's suggestion and we have added the definition of polypharmacy in line 84.
Point 3. Lines 100-102: Need to be more specific about why existing studies cannot be extrapolated to the current situation.
Response 3. The studies referred to are references 16, 26-28. They are studies with the same objective as ours, but they are from different countries, which implies different prescribing habits and the use of different prescribing systems. In addition, study 16 recruited patients admitted to a medical or surgical unit, not to EDs like ours. And in studies 25 and 26 the patient population differs from ours in two characteristics relevant to this study: they are patients aged 50 or older and using 5 or more drugs, while in our case they are 18 or older and we have patients using 2 or more drugs.
A sentence has been added to line 104 to clarify this in the manuscript.
Discussion
Point 4. Line 193: Don’t need “which report medians of 2 to 3 discrepancies per patient” as it was just noted two sentences prior and the subject did not change.
Response 4. We appreciate the reviewer's suggestion and have made the proposed change.
Point 5. Line 196: Specify what “The same reason” is.
Response 5. We appreciate the reviewer's suggestion and to clarify the context of the sentence we have modified line 195.
Materials and Methods
Point 6. Line 327: Change “damage” to “harm” to be more consistent with common patient safety language.
Response 6. We appreciate the reviewer's suggestion and have made the proposed change.

Reviewer 3 Report
Comments and Suggestions for Authors
The authors revised their paper in accordance with my previous comments. They accepted all of my proposed recommendations and therefore I suggest an acceptance.
Author Response
General comments. The authors revised their paper in accordance with my previous comments. They accepted all of my proposed recommendations and therefore I suggest an acceptance.
Response to general comments. We appreciate all the improvements suggested by the reviewer during the first round. We are pleased that all comments were duly answered and the manuscript improved.
